# Histone Deacetylase Inhibitory Activity and Antiproliferative Potential of New [6]-Shogaol Derivatives

**DOI:** 10.3390/molecules27103332

**Published:** 2022-05-22

**Authors:** Chanokbhorn Phaosiri, Chavi Yenjai, Thanaset Senawong, Gulsiri Senawong, Somprasong Saenglee, La-or Somsakeesit, Pakit Kumboonma

**Affiliations:** 1Natural Products Research Unit, Center of Excellence for Innovation in Chemistry, Ministry of Higher Education, Science, Research and Innovation (Implementation Unit-IU, Khon Kaen University), Department of Chemistry, Faculty of Science, Khon Kaen University, Khon Kaen 40002, Thailand; chapha@kku.ac.th (C.P.); chayen@kku.ac.th (C.Y.); 2Natural Products Research Unit, Department of Biochemistry, Faculty of Science, Khon Kaen University, Khon Kaen 40002, Thailand; sthanaset@kku.ac.th (T.S.); gulsiri@kku.ac.th (G.S.); 3Ban Dong Sub-District Administration Organization, Ubolratana District, Khon Kaen 40250, Thailand; bird_scorpio@hotmail.com; 4Department of Chemistry, Faculty of Engineering, Rajamangala University of Technology Isan, Khon Kaen 40000, Thailand; laor.so@rmuti.ac.th; 5Department of Applied Chemistry, Faculty of Science and Liberal Arts, Rajamangala University of Technology Isan, Nakhon Ratchasima 30000, Thailand

**Keywords:** ginger, *Zingiber officinale*, [6]-shogaol derivatives, anticancer, molecular docking

## Abstract

Twenty newly synthesized derivatives of [6]-shogaol (**4**) were tested for inhibitory activity against histone deacetylases. All derivatives showed moderate to good histone deacetylase inhibition at 100 µM with a slightly lower potency than the lead compound. Most potent inhibitors among the derivatives were the pyrazole products, **5j** and **5k,** and the Michael adduct with pyridine **4c** and benzothiazole **4d,** with IC_50_ values of 51, 65, 61 and 60 µM, respectively. They were further evaluated for isoform selectivity via a molecular docking study. Compound **4d** showed the best selectivity towards HDAC3, whereas compound **5k** showed the best selectivity towards HDAC2. The potential derivatives were tested on five cancer cell lines, including human cervical cancer (HeLa), human colon cancer (HCT116), human breast adenocarcinoma cancer (MCF-7), and cholangiocarcinoma (KKU100 and KKU-M213B) cells with MTT-based assay. The most active histone deacetylase inhibitor **5j** exhibited the best antiproliferative activity against HeLa, HCT116, and MCF-7, with IC_50_ values of 8.09, 9.65 and 11.57 µM, respectively, and a selective binding to HDAC1 based on molecular docking experiments. The results suggest that these compounds can be putative candidates for the development of anticancer drugs via inhibiting HDACs.

## 1. Introduction

The acetylation of histones by histone acetyltransferases (HATs) and the deacetylation of histones by histone deacetylases (HDACs) support the epigenetic regulation of gene expression to a substantial degree [1,2]. Epigenetic changes play an important role in tumorigenesis. Epigenetic post-translational modifications such as lysine deacetylation are involved in the progression of cancer [3]. HDACs have key effects on various cellular functions such as the regulation of gene transcription and cell proliferation, differentiation and death [4,5]. Moreover, HDACs are overexpressed in many human cancer cells. Therefore, HDAC inhibitors have been extensively explored as epigenetic therapeutics for cancer [6,7,8,9,10]. Currently, four HDAC inhibitors, including vorinostat (**1**, Zolinza^®^), romidepsin (Istodax^®^), belinostat (Beleodaq^®^) and panobinostat (Farydak^®^), have been approved by the U.S. Food and Drug Administration (USFDA) and launched on the market for the treatment of multiple myloma [11,12,13,14]. In addition, another HDAC inhibitor, chidamide (Epidaza^®^), has been approved by the Chinese Food and Drug Administration (CFDA) for the treatment of hematologic cancer [15]. There are eighteen known HDACs, grouped into four classes based on sequence similarity [16]. Class I (HDAC1, 2, 3 and 8), class IIa (HDAC4, 5, 7 and 9), class IIb (HDAC6 and 10) and class IV (HDAC11) HDACs are zinc-dependent enzymes, whereas class III HDACs (SIRT1-SIRT7) require NAD^+^ for activity [17]. All four USFDA-approved HDAC inhibitors are hydroxamic acids and pan-inhibitors that non-selectively inhibit most of HDAC isoforms (HDAC1-11) [18,19]. This non-selective property might explain the side effects observed in clinic and then limit their use in cancer therapy [20]. Moreover, hydroxamic acid, a strong zinc chelator, shows metabolic and pharmacokinetic issues, including glucuronidation, sulfonation and enzymatic hydrolysis, that result in a short in vivo half-life [21]. Therefore, the search for developing potent HDAC inhibitors with isoform selectivity and minimum side effects continues [22,23,24,25]. Several non-hydroxamic HDAC inhibitors have been reported, such as hydroxycapsaicin (**2**), [6]-gingerol (**3**) and [6]-shogaol (**4**) (Figure 1) [26,27,28,29,30]. Even though these compounds possess less HDAC inhibitory potency than hydroxamic HDAC inhibitors, they caught our attention as natural-derived compounds with low toxicities [31]. Natural-derived [6]-shogaol (**4**) had received a lot of attention because of its variety of biological activities, such as anti-inflammatory, antioxidant, anticancer, antibacterial activities, and HDAC inhibitor [32]. In this study, we continued with the modifications of [6]-shogaol (**4**) to improve inhibitor enzyme binding and aimed to determine whether these compounds would be potent HDAC inhibitors and manifest antiproliferative activities against cancer cells, as well as maintaining their low toxicity. The [6]-shogaol derivatives were designed to increase inhibitor enzyme binding via van der Waals, hydrogen bonds, hydrophobic and hydrophilic bonds, and π–π interactions by introducing hydrazone, pyrazole, amino and imine moiety into the molecular skeleton of [6]-shogaol (**4**).

## 2. Results and Discussion

The natural compound [6]-shogaol (**4**) was isolated from ginger as previously described [27]. The immino derivatives **4a**–**4e** were prepared by reacting [6]-shogaol (**4**) with hydrazine hydrate, 4-hydrazinobenzoic acid, 2-hydrazinopyridine, 2-hydrazinobenzothiazole and 2-hydrazino-2-imidazoline in ethanol at room temperature (Figure 1). Reactions of [6]-shogaol (**4**) with phenylhydrazines including phenylhydrazine, *o*-tolylphenylhydrazine, 2-methoxyphenylhydrazine, 4-methoxyphenylhydrazine, 2-fluorophenylhydrazine, 4-fluorophenylhydrazine, 3-chlorophenylhydrazine, 4-chlorophenylhydrazine, 3-nitrophenylhydrazine, 4-nitrophenylhydrazine, and 4-hydrazinobenzoic acid in ethanol at 80 °C provided the pyrazole derivatives **5a**–**5k**. All derivatives were obtained in good yields. The formations of the pyrazole rings were confirmed by the ^13^C-NMR chemical shift of about 62 and 153 ppm for the C-6 and C-8 positions, respectively. Reactions of [6]-shogaol (**4**) with primary amines, including 2-aminothiophenol, 2-aminophenol and aniline, in ethanol at 80 °C provided the Michael-addition products **6a**–**6c**. Interestingly, Michael immino adduct **7** was gained as the major product from reacting [6]-shogaol (**4**) with *o*-phenylenediamine in ethanol at 80 °C. The reaction showed that the amino group at the *ortho*-position reacts at the carbonyl group to provide a seven-membered ring. The formation of the seven-membered ring of **7** was confirmed by the chemical shift of ^13^C-NMR at 174.1 and 65.7 ppm.

The synthesized derivatives were tested for HDAC inhibition with a commercial HDAC assay kit. The results are summarized in Table 1. All derivatives showed slightly weaker % HDAC inhibitions than the lead compound. The immino derivatives **4c** and **4d** showed the best HDAC inhibitory activity among the immino derivatives, with IC_50_ values of 61 ± 0.92 and 60 ± 0.84 µM, respectively. The immino derivatives with aromatic groups **4a**–**4e** showed stronger HDAC inhibitory activities than the Michael-addition products **6a**–**6c**, **7**. However, this Michael-adduct type of [6]-shogaol could become thymidylate kinase inhibitors [33].

To study the structure–activity relationship (SAR), phenyl pyrazole derivatives with various substitution groups at the aromatic region, such as methyl, methoxy, fluoro, chloro, nitro, and carboxyl groups, were evaluated as HDAC inhibitors. The substitution phenylpyrazoles **5b**–**5h** showed slightly weaker % HDAC inhibition than the phenylpyrazole **5a**. Interestingly, the *para*-substitution derivatives showed highly potent HDAC inhibitors than *ortho*- and *meta*-substitution derivatives. The *p*-nitrophenyl, pyrazole **5j,** and *p*-carboxylphenyl pyrazole **5k** derivatives were the most potent inhibitors among the pyrazole derivatives with IC_50_ values of 51 ± 0.82 and 65 ± 1.12 µM, respectively.

The most potent derivatives, **4c**, **4d**, **5j** and **5k,** were further investigated as the isoform-selective inhibitors of the isoforms class I HDACs (HDAC1, HDAC2, HDAC3 and HDAC8). The results from the molecular docking experiment are shown in Table 2. The immino derivative **4d** showed a better binding affinity with HDAC1 and HDAC3 isoforms than TSA. The major interaction of **4d** and HDAC1 (Figure 2) consisted of two hydrogen bonds between **4d** and Phe205 (2.9 Å) and Leu271 (2.7 Å). The side chain of **4d** was inserted into the catalytic channel of HDAC1, binding the cofactor Zn^2+^ ion. The binding mode of **4d** and HDAC3 is shown in Figure 3. The **4d**-HDAC3 complex showed that a stronger inhibitor–enzyme interaction consists in two hydrogen bonds between **4d** and Asp92 (2.1 Å) and Gly143 (3.3 Å) of HDAC3. The coordination of the benzotriazole group to the Zn^2+^ ion (3.2 Å) can also be observed. Compound **5k** showed a selectivity to HDAC2. The **5k**-HDAC2 complex showed a complete insertion of its aromatic ring into the active site pocket, with multiple contacts with the tubular channel. The major interaction of **5k** and HDAC2 (Figure 4) consisted of three hydrogen bonds between **5k** and Gly154 (2.7 Å), Gly143 (2.7 Å), Leu276 (3.1 Å), the π–π interaction with Phe210, as well as the coordination of the aromatic ring to the Zn^2+^ ion (2.8 Å). The most potent compound in vitro **5j** showed selectivity to HDAC1. The **5j**-HDAC1 complex shows that the stronger inhibitor enzyme interaction consists of three hydrogen bonds between **5j** and Phe150 (2.7 Å), Cys151 (2.7 Å), Gly300 (2.8 Å), the π–π interactions with Phe150, His140, and His178; as well as the coordination of the aromatic ring to the Zn^2+^ ion (2.0 Å) (Figure 5).

The in vitro HDAC inhibitions of the obtained compounds were carried out with the assay kit containing mixed-HDAC isoforms. In order to further investigate the isoform selectivity of the four most potent HDAC inhibitors, an in silico experiment was performed with each HDAC isoform. Therefore, the results were not fully related. However, compounds with good binding affinities should have the potential to be specific HDAC inhibitors for each isoform.

To complete the evaluation of these potent HDAC inhibitors, antiproliferative activities were determined in human cervical cancer (HeLa), human colon cancer (HCT116), human breast adenocarcinoma cancer (MCF-7), and cholangiocarcinoma (KKU-100 and KKU-M213B) cells. The results, as shown in Table 3 and Table 4, indicate that the derivatives **4c**, **4d**, **5j** and **5k** were less toxic to non-cancer cells than the lead compound. Compound **5j** showed the best activity against HeLa, HCT116 and MCF-7, with IC_50_ values of 8.09, 9.65 and 11.57 µM, respectively. Additionally, compound **5j** displayed a strong antiproliferative activity against cholangiocarcinoma cells, as shown in Table 4. Compounds **4c** and **4d** exhibited antiproliferative activity against cholangiocarcinoma cells with selectivity towards KKU-100 and KKU-M213B, respectively. Compound **5k**, the least toxic compound, showed a good activity against HeLa cells with an IC_50_ value of 23.14 µM. Moreover, compound **5k** appeared to be about 5-fold more selective towards HeLa cells than non-cancer cells, whereas [6]-shogaol (**4**) exhibited only a 3-fold selectivity.

Anticancer activities of the best-four HDAC inhibitors were investigated upon the proliferation of five human cancer cell lines in a time-dependent manner. All selected compounds showed an antiproliferative activity in the growth inhibition of cancer cell lines. The results support the use of these HDAC inhibitors as potential anticancer candidates.

## 3. Conclusions

In conclusion, a series of new immino and pyrazole derivatives of [6]-shogaol were designed and synthesized as potential HDACs inhibitors. All derivatives exhibited HDAC inhibitory activity in the micromolar concentration ranges. Among these derivatives, pyrazole **5j** with *p*-nitro substituent displayed the most remarkable HDACs inhibitory activities. Additionally, pyrazole **5j** showed the best activity against all tested cancer cells among the synthesized derivatives and the most selective binding to HDAC1 based on molecular docking experiments. Therefore, the *in vivo* experiments of **5j** will be further investigated.

## 4. Experimental Section

### 4.1. General

Reagents were purchased from commercial sources (Sigma-Aldrich, Merck, and Carlo Erba). Reactions were monitored using analytical TLC plates (Merck, silica gel 60 F_254_), and compounds were visualized under ultraviolet light. Silica gel grade 60 (230–400 mesh, Merck) was used for column chromatography. The NMR spectra were recorded in the indicated solvents on a Varian Mercury Plus spectrometer operated at 400 MHz (^1^H) or 100 MHz (^13^C). The IR spectra were obtained on Perkin Elmer Spectrum One FT-IR spectrophotometer. Mass spectra were determined using a Micromass Q-TOF 2 hybrid quadrupole time-of-flight (Q-TOF) mass spectrometer with a Z-spray ES source.

### 4.2. Plant Material

The dried rhizome powder of ginger was obtained from the local market in Khon Kaen province, Thailand. The extraction and isolation of [6]-shogaol (**4**) followed previously described methods [27].

### 4.3. Structural Modifications

#### 4.3.1. Synthesis of the Immino Derivatives (**4a**–**4e**)

To a solution of [6]-shogaol (**4**) (113 mg, 0.41 mmol) in ethanol (5 mL), NH_2_NH_2_.2HCl (64 mg, 0.76 mmol) was added. The solution was stirred at room temperature until completion based on TLC. The mixture was filtered, and the residue was washed with ethanol (10 mL) and dried with anhydrous sodium sulfate. Evaporation of the combined solvents gave a crude product. Purification of the crude product by column chromatography (100% dichloromethane) gave the yellow oil of compound **4a** (90 mg, 0.31 mmol, 76%).

4-((3*Z*,4*E*)-3-Hydrazonodec-4-en-1-yl)-2-methoxyphenol (**4a**). ^1^H NMR spectrum of **4a** is shown in Appendix A. Yellow oil; yield: 76%; R*_f_* = 0.50 (100% dichlorometane); IR (neat) *υ*_max_ 3336 (OH), 1604 (C=N), 1515 (C=C), 1278 (C-O) cm^−1^.^1^H NMR (CDCl_3_, 400 MHz) *δ* 6.82 (dd, *J* = 2.0, 8.0 Hz, 1H), 6.69 (m, 2H), 6.09 (m, 1H), 5.48 (s, 1H), 3.87 (s, 3H), 2.83 (m, 2H), 2.75 (m, 2H), 2.69 (t, *J* = 8.0 Hz, 1H), 2.60 (q, *J* = 8.0 Hz, 1H), 1.42 (m, 2H), 1.25 (m, 4H), 0.87 (m, 3H). ^13^C NMR (CDCl_3_, 100 MHz) *δ* 149.1, 147.9, 146.4, 143.8, 133.2, 130.3, 120.8, 114.3, 111.0, 55.9, 42.0, 32.5, 29.2, 28.1, 23.8, 22.5, 14.1. HRMS-ESI (m/z) [M + Na]^+^ calcd for C_17_H_26_N_2_O_2_Na 313.1892, found 313.1882.

The same procedure was carried out with 4-hydrazinobenzoic acid hydrochloride, 2-hydrazinopyridine, 2-hydrazinobenzothiazole, and 2-hydrazino-2-imidazoline to convert [6]-shogaol (**4**) into **4b**–**4e**.

4-(2-((3*Z*,4*E*)-1-(4-Hydroxy-3-methoxyphenyl)dec-4-en-3-ylidene)hydrazinyl)benzo-ic acid (**4b**). Yellow viscous liquid; yield: 75%; R*_f_* = 0.45 (100% dichlorometane); IR (neat) *υ*_max_ 3336 (OH), 1692 (C=O), 1603 (C=N), 1546 (C=C), 1268 (C-O) cm^−1^. ^1^H NMR (CDCl_3_, 400 MHz) *δ* 8.07 (d, *J* = 8.0 Hz, 2H), 7.25 (d, *J* = 8.0 Hz, 2H), 6.79 (d, *J* = 8.0 Hz, 1H), 6.62 (d, *J* = 8.0 Hz, 1H), 6.45 (d, *J* = 4.0 Hz, 1H), 6.40 (dd, *J* = 4.0, 8.0 Hz, 1H), 6.11 (s, 1H), 3.69 (s, 3H), 2.73 (t, *J* = 8.0 Hz, 2H), 2.59 (t, *J* = 8.0 Hz, 2H), 1.64 (m, 2H), 1.34 (m, 4H), 0.90 (m, 3H). ^13^C NMR (CDCl_3_, 100 MHz) *δ* 161.5, 154.2, 147.4, 144.5, 144.4, 141.9, 131.9, 130.2, 124.9, 120.5, 114.7, 114.6, 111.6, 105.1, 55.1, 34.8, 31.3, 29.1, 28.0, 27.6, 22.2, 13.2. HRMS-ESI (m/z) [M − H]^+^ calcd for C_24_H_29_N_2_O_4_ 409.2127, found 409.2198.

2-Methoxy-4-((3*Z*,4*E*)-3-(2-(pyridin-2-yl)hydrazono)dec-4-en-1-yl)phenol (**4c**). Orange viscous liquid; yield: 70%; R*_f_* = 0.65 (100% dichlorometane); IR (neat) *υ*_max_ 3332 (OH), 1589 (C=N), 1513 (C=C), 1268 (C-O) cm^−1^.^1^H NMR (CDCl_3_, 400 MHz) *δ* 8.42 (m, 1H), 7.78 (m, 2H), 7.15 (m, 1H), 6.82 (m, 1H), 6.75 (m, 1H), 6.68 (s, 1H), 6.03 (s, 1H), 5.54 (brs, 1H), 3.85 (s, 3H), 3.37 (t, *J* = 8.0 Hz, 2H), 2.91 (m, 2H), 2.63 (t, *J* = 8.0 Hz, 2H), 1.64 (m, 2H), 1.36 (m, 4H), 0.98 (m, 3H). ^13^C NMR (CDCl_3_, 100 MHz) *δ* 154.6, 153.6, 147.4, 146.3, 145.2, 143.8, 138.2, 133.5, 121.0, 120.7, 116.3, 114.2, 111.0, 107.0, 55.8, 35.2, 31.7, 30.1, 29.2, 28.3, 22.5, 14.0. HRMS-ESI (m/z) [M + H]^+^ calcd for C_22_H_30_N_3_O_2_ 368.2338, found 368.2356.

4-((3*Z*,4*E*)-3-(2-(Benzo[d]thiazol-2-yl)hydrazono)dec-4-en-1-yl)-2-methoxyphenol (**4d**). Yellow viscous liquid; yield: 78%; R*_f_* = 0.40 (100% dichlorometane); IR (neat) *υ*_max_ 3336 (OH), 1600 (C=N), 1513 (C=C), 1267 (C-O) cm^−1^.^1^H NMR (CDCl_3_, 400 MHz) *δ* 7.56 (d, *J* = 8.0 Hz, 2H), 7.40 (d, *J* = 8.0 Hz, 2H), 7.24 (t, *J* = 8.0 Hz, 1H), 7.06 (t, *J* = 8.0 Hz, 1H), 6.75 (d, *J* = 8.0 Hz, 1H), 6.73 (s, 1H), 6.67 (dd, *J* = 4.0, 8.0 Hz, 1H), 6.14 (d, *J* = 16.0 Hz, 1H), 6.02 (m, 1H), 3.80 (s, 3H), 2.71 (brs, 4H), 2.14 (q, *J* = 8.0 Hz, 2H), 1.39 (m, 2H), 1.25 (m, 4H), 0.84 (m, 3H). ^13^C NMR (CDCl_3_, 100 MHz) *δ* 168.5, 153.6, 149.4, 147.0, 144.3, 136.4, 132.4, 129.7, 129.3, 125.9, 121.9, 121.2, 121.0, 117.8, 114.9, 111.6, 55.8, 32.9, 31.8, 31.3, 29.0, 28.1, 22.4, 13.9. HRMS-ESI (m/z) [M + H]^+^ calcd for C_24_H_30_N_3_O_2_S 424.2059, found 424.2212.

4-((3*Z*,4*E*)-3-(2-(4*H*-Imidazol-2-yl)hydrazono)dec-4-en-1-yl)-2-methoxyphenol (**4e**). Yellow viscous liquid; yield: 72%; R*_f_* = 0.45 (100% dichlorometane); IR (neat) *υ*_max_ 3158 (OH), 1657 (C=N), 1611 (C=N), 1513 (C=C), 1274 (C-O) cm^−1^.^1^H NMR (CDCl_3_, 400 MHz) *δ* 6.81 (s, 1H), 6.71 (m, 1H), 6.64 (m, 1H), 6.01 (m, 2H), 3.82 (s, 3H), 3.59 (brs, 4H), 2.77 (m, 2H), 2.64 (m, 2H), 2.11 (q, *J* = 8.0 Hz, 2H), 1.36 (m, 2H), 1.24 (m, 4H), 0.83 (m, 3H). ^13^C NMR (CDCl_3_, 100 MHz) *δ* 161.2, 158.2, 147.0, 144.1, 137.1, 132.8, 129.4, 120.8, 114.5, 111.8, 55.8, 42.6, 32.9, 32.3, 31.8, 29.0, 28.4, 22.4, 13.9. HRMS-ESI (m/z) [M + H]^+^ calcd for C_20_H_31_N_4_O_2_ 359.2447, found 359.2443.

#### 4.3.2. Synthesis of the Pyrazole Derivatives (**5a**–**5k**)

To a solution of [6]-shogaol (**4**) (207 mg, 0.75 mmol) in ethanol (5 mL), phenylhydrazine hydrochloride (108 mg, 0.75 mmol) was added. The solution was refluxed at 80 °C until completion based on TLC. The mixture was filtered, and the residue was washed with ethanol (10 mL) and dried with anhydrous sodium sulfate. Evaporation of the combined solvents gave a crude product. Purification of the crude product by column chromatography (10% ethyl acetate/hexane) gave a dark green viscous liquid of compound **5a** (212 mg, 0.58 mmol, 77%).

2-methoxy-4-(2-(5-pentyl-1-phenyl-4,5-dihydro-1*H*-pyrazol-3-yl)ethyl)phenol (**5a**). Dark green viscous liquid; yield: 77%; R*_f_* = 0.65 (70% dichlorometane: hexane); IR (neat) *υ*_max_ 3336 (OH), 1597 (C=N), 1514 (C=C), 1269 (C-O), 1119 (C-N), 1033 (C-O) cm^−1^.^1^H NMR (CDCl_3_, 400 MHz) *δ* 7.40 (t, *J* = 8.0 Hz, 2H), 7.18 (d, *J* = 8.0 Hz, 2H), 6.97 (d, *J* = 8.0 Hz, 1H), 6.90 (m, 3H), 5.75 (s, 1H), 4.23 (m, 1H), 3.98 (s, 3H), 3.12 (dd, *J* = 4.0, 8.0 Hz, 1H), 3.02 (t, *J* = 8.0 Hz, 2H), 2.82 (t, *J* = 8.0 Hz, 2H), 2.65 (dd, *J* = 4.0, 8.0 Hz, 1H), 1.90 (m, 1H), 1.55 (m, 1H), 1.45 (m, 6H), 1.02 (t, *J* = 8.0 Hz, 3H). ^13^C NMR (CDCl_3_, 100 MHz) *δ* 152.0, 146.3, 145.6, 143.8, 133.1, 128.9, 120.8, 118.2, 114.2, 113.1, 111.0, 59.8, 55.7, 41.1, 32.6, 32.5, 32.4, 31.6, 24.7, 22.6, 13.9. HRMS-ESI (m/z) [M − H]^+^ calcd for C_23_H_29_N_2_O_2_ 365.2229, found 365.2222.

The same procedure was carried out with *o*-tolylphenylhydrazine hydrochloride, 2-methoxyphenylhydrazine hydrochloride, 4-methoxyphenylhydrazine hydrochloride, 2-fluorophenylhydrazine hydrochloride, 4-fluorophenylhydrazine hydrochloride, 3-chlorophenylhydrazine hydrochloride, 4-chlorophenylhydrazine hydrochloride, 3-nitrophenylhydrazine hydrochloride, 4-nitrophenylhydrazine hydrochloride, and 4-hydrazinobenzoic acid to convert [6]-shogaol (**4**) into **5b**–**5k**.

2-methoxy-4-(2-(5-pentyl-1-(*o*-tolyl)-4,5-dihydro-1*H*-pyrazol-3-yl)ethyl)phenol (**5b**). Brown viscous liquid; yield: 80%; R*_f_* = 0.48 (70% dichlorometane: hexane); IR (neat) *υ*_max_ 3336 (OH), 1598 (C=N), 1514 (C=C), 1269 (C-O), 1119 (C-N), 1033 (C-O) cm^−1^.^1^H NMR (CDCl_3_, 400 MHz) *δ* 7.25 (m, 3H), 7.10 (t, *J* = 8.0 Hz, 1H), 6.92 (d, *J* = 8.0 Hz, 1H), 6.82 (m, 2H), 5.80 (s, 1H), 4.00 (qd, *J* = 4.0, 8.0 Hz, 1H), 3.92 (s, 3H), 2.95 (m, 3H), 2.75 (t, *J* = 8.0 Hz, 2H), 2.58 (dd, *J* = 4.0, 8.0 Hz, 1H), 2.45 (s, 3H), 1.30 (m, 8H), 0.95 (t, *J* = 8.0 Hz, 3H). ^13^C NMR (CDCl_3_, 100 MHz) *δ* 153.2, 146.3, 145.5, 143.8, 133.1, 131.9, 130.7, 126.1, 123.7, 121.2, 120.7, 114.2, 111.0, 64.4, 55.7, 40.7, 32.6, 32.5, 31.6, 31.4, 25.7, 22.5, 19.1, 13.8. HRMS-ESI (m/z) [M − H]^+^ calcd for C_24_H_31_N_2_O_2_ 379.2386, found 379.2353.

2-methoxy-4-(2-(1-(2-methoxyphenyl)-5-pentyl-4,5-dihydro-1*H*-pyrazol-3-yl)ethyl) phenol (**5c**). Brown viscous liquid; yield: 82%; R*_f_* = 0.40 (70% dichlorometane: hexane); IR (neat) *υ*_max_ 3330 (OH), 1594 (C=N), 1513 (C=C), 1232 (C-O), 1121 (C-N), 1036 (C-O) cm^−1^.^1^H NMR (CDCl_3_, 400 MHz) *δ* 7.20 (d, *J* = 8.0 Hz, 1H), 6.85 (m, 1H), 6.75 (m, 3H), 6.60 (m, 2H), 5.45 (s, 1H), 4.25 (m, 1H), 3.75 (s, 3H), 2.75 (m, 3H), 2.55 (t, *J* = 8.0 Hz, 2H), 2.35 (dd, *J* = 4.0, 8.0 Hz, 1H), 1.10 (m, 8H), 0.70 (t, *J* = 8.0 Hz, 3H). ^13^C NMR (CDCl_3_, 100 MHz) *δ* 153.0, 150.2, 146.3, 143.8, 135.7, 133.2, 122.8, 121.1, 121.0, 120.8, 114.2, 111.0, 110.9, 62.8, 55.8, 55.4, 40.4, 32.8, 32.5, 31.5, 30.8, 25.0, 22.4, 13.9. HRMS-ESI (m/z) [M − H]^+^ calcd for C_24_H_31_N_2_O_3_ 395.2335, found 395.2306.

2-methoxy-4-(2-(1-(4-methoxyphenyl)-5-pentyl-4,5-dihydro-1*H*-pyrazol-3-yl)ethyl) phenol (**5d**). Brown viscous liquid; yield: 79%; R*_f_* = 0.40 (70% dichlorometane: hexane); IR (neat) *υ*_max_ 3336 (OH), 1607 (C=N), 1512 (C=C), 1247 (C-O), 1120 (C-N), 1031 (C-O) cm^−1^.^1^H NMR (CDCl_3_, 400 MHz) *δ* 7.20 (d, *J* = 8.0 Hz, 2H), 7.05 (m, 3H), 6.95 (m, 2H), 5.78 (s, 1H), 4.10 (s, 1H), 4.08 (s, 3H), 4.05 (s, 3H), 3.98 (s, 3H), 3.15 (dd, *J* = 4.0, 8.0 Hz, 1H), 3.08 (t, *J* = 8.0 Hz, 2H), 2.85 (t, *J* = 8.0 Hz, 2H), 2.70 (dd, *J* = 4.0, 8.0 Hz, 1H), 1.95 (m, 2H), 1.50 (m, 6H), 1.10 (t, *J* = 8.0 Hz, 3H). ^13^C NMR (CDCl_3_, 100 MHz) *δ* 153.4, 152.2, 146.4, 143.8, 140.7, 133.2, 120.8, 116.1, 114.4, 114.2, 111.0, 62.2, 55.8, 55.6, 41.3, 32.7, 32.7, 32.5, 31.7, 25.2, 22.6, 14.0. HRMS-ESI (m/z) [M – H]^+^ calcd for C_24_H_31_N_2_O_3_ 395.2335, found 395.2307.

4-(2-(1-(2-fluorophenyl)-5-pentyl-4,5-dihydro-1*H*-pyrazol-3-yl)ethyl)-2-methoxyphenol (**5e**). Brown viscous liquid; yield: 82%; R*_f_* = 0.55 (70% dichlorometane: hexane); IR (neat) *υ*_max_ 3336 (OH), 1610 (C=N), 1514 (C=C), 1269 (C-O), 1233 (C-F), 1118 (C-N), 1033 (C-O) cm^−1^. ^1^H NMR (CDCl_3_, 400 MHz) *δ* 7.60 (t, *J* = 8.0 Hz, 1H), 7.18 (m, 2H), 7.00 (m, 2H), 6.90 (m, 2H), 5.90 (s, 1H), 4.23 (m, 1H), 3.98 (s, 3H), 3.08 (m, 1H), 3.05 (t, *J* = 8.0 Hz, 2H), 2.85 (t, *J* = 8.0 Hz, 2H), 2.68 (dd, *J* = 4.0, 8.0 Hz, 1H), 1.55 (m, 1H), 1.40 (m, 7H), 1.00 (t, *J* = 8.0 Hz, 3H). ^13^C NMR (CDCl_3_, 100 MHz) *δ* 153.4, 150.7, 146.4, 143.8, 134.0, 132.9, 124.3, 121.3, 120.7, 120.1, 115.7, 114.2, 110.9, 62.3, 55.7, 40.5, 32.6, 32.2, 31.4, 31.2, 24.6, 22.4, 13.8. HRMS-ESI (m/z) [M − H]^+^ calcd for C_23_H_28_FN_2_O_2_ 383.2135, found 383.2121.

4-(2-(1-(4-fluorophenyl)-5-pentyl-4,5-dihydro-1*H*-pyrazol-3-yl)ethyl)-2-methoxyphenol (**5f**). Brown viscous liquid; yield: 80%; R*_f_* = 0.55 (70% dichlorometane: hexane); IR (neat) *υ*_max_ 3336 (OH), 1605 (C=N), 1506 (C=C), 1269 (C-O), 1219 (C-F), 1120 (C-N), 1033 (C-O) cm^−1^. ^1^H NMR (CDCl_3_, 400 MHz) *δ* 7.18 (m, 4H), 7.05 (d, *J* = 8.0 Hz, 1H), 6.95 (m, 2H), 5.90 (s, 1H), 4.20 (m, 1H), 4.05 (s, 3H), 3.18 (m, 1H), 3.10 (t, *J* = 8.0 Hz, 2H), 2.88 (t, *J* = 8.0 Hz, 2H), 2.72 (dd, *J* = 4.0, 8.0 Hz, 1H), 1.95 (m, 1H), 1.65 (m, 1H), 1.50 (m, 6H), 1.10 (t, *J* = 8.0 Hz, 3H). ^13^C NMR (CDCl_3_, 100 MHz) *δ* 152.4, 146.4, 143.8, 133.0, 120.8, 115.4, 115.2, 114.8, 114.7, 114.2, 111.0, 60.9, 55.7, 41.2, 32.6, 32.4, 32.3, 31.6, 24.9, 22.5, 13.9. HRMS-ESI (m/z) [M − H]^+^ calcd for C_23_H_28_ FN_2_O_2_ 383.2135, found 383.2122.

4-(2-(1-(3-chlorophenyl)-5-pentyl-4,5-dihydro-1*H*-pyrazol-3-yl)ethyl)-2-methoxyphenol (**5g**). Dark viscous liquid; yield: 75%; R*_f_* = 0.60 (70% dichlorometane: hexane); IR (neat) *υ*_max_ 3336 (OH), 1592 (C=N), 1513 (C=C), 1269 (C-O), 1233 (C-N), 1033 (C-O) cm^−1^. ^1^H NMR (CDCl_3_, 400 MHz) *δ* 7.25 (t, *J* = 8.0 Hz, 1H), 7.20 (d, *J* = 2.0 Hz, 1H), 6.98 (d, *J* = 8.0 Hz, 1H), 6.95 (dd, *J* = 2.0, 8.0 Hz, 1H), 6.90 (d, *J* = 2.0 Hz, 1H), 6.85 (dd, *J* = 2.0, 8.0 Hz, 2H), 5.70 (s, 1H), 4.20 (m, 1H), 3.98 (s, 3H), 3.10 (dd, *J* = 4.0, 8.0 Hz, 1H), 3.00 (t, *J* = 8.0 Hz, 2H), 2.80 (t, *J* = 8.0 Hz, 2H), 2.62 (dd, *J* = 4.0, 8.0 Hz, 1H), 1.85 (m, 1H), 1.55 (m, 1H), 1.40 (m, 6H), 1.00 (t, *J* = 8.0 Hz, 3H). ^13^C NMR (CDCl_3_, 100 MHz) *δ* 152.8, 146.3, 146.3, 143.8, 134.7, 132.9, 129.9, 120.7, 117.6, 114.2, 112.7, 111.0, 110.5, 59.3, 55.7, 41.1, 32.4, 32.2, 32.1, 31.5, 24.4, 22.5, 13.9. HRMS-ESI (m/z) [M − H]^+^ calcd for C_23_H_28_ClN_2_O_2_ 399.1839, found 399.1802.

4-(2-(1-(4-chlorophenyl)-5-pentyl-4,5-dihydro-1H-pyrazol-3-yl)ethyl)-2-methoxyphenol (**5h**). Dark viscous liquid; yield: 83%; R*_f_* = 0.62 (70% dichlorometane: hexane); IR (neat) *υ*_max_ 3336 (OH), 1596 (C=N), 1514 (C=C), 1269 (C-O), 1233 (C-N), 1033 (C-O), 816 (C-Cl) cm^−1^. ^1^H NMR (CDCl_3_, 400 MHz) *δ* 7.20 (d, *J* = 8.0 Hz, 2H), 6.95 (d, *J* = 8.0 Hz, 2H), 6.88 (d, *J* = 8.0 Hz, 1H), 6.78 (d, *J* = 2.0 Hz, 1H), 6.75 (d, *J* = 2.0, 8.0 Hz, 1H), 5.65 (s, 1H), 4.05 (m, 1H), 3.85 (s, 3H), 3.00 (dd, *J* = 4.0, 8.0 Hz, 1H), 2.90 (t, *J* = 8.0 Hz, 2H), 2.68 (t, *J* = 8.0 Hz, 2H), 2.52 (dd, *J* = 2.0, 8.0 Hz, 1H), 1.75 (m, 1H), 1.40 (m, 1H), 1.30 (m, 6H), 0.90 (t, *J* = 8.0 Hz, 3H). ^13^C NMR (CDCl_3_, 100 MHz) *δ* 152.5, 146.3, 144.0, 143.8, 133.0, 128.7, 122.7, 120.8, 114.2, 114.0, 110.9, 59.6, 55.7, 41.2, 32.5, 32.2, 32.2, 31.5, 24.6, 22.5, 13.9. HRMS-ESI (m/z) [M − H]^+^ calcd for C_23_H_28_ClN_2_O_2_ 399.1839, found 399.1809.

2-methoxy-4-(2-(1-(3-nitrophenyl)-5-pentyl-4,5-dihydro-1*H*-pyrazol-3-yl)ethyl)phe nol (**5i**). Brown viscous liquid; yield: 70%; R*_f_* = 0.35 (70% dichlorometane: hexane); IR (neat) *υ*_max_ 3508 (OH), 1614 (C=N), 1514 (C=C), 1491 (N-O), 1463 (Ar), 1269 (C-O), 1233 (C-N), 1032 (C-O) cm^−1^. ^1^H NMR (CDCl_3_, 400 MHz) *δ* 7.70 (s, 1H), 7.50 (dd, *J* = 4.0, 8.0 Hz, 1H), 7.25 (t, *J* = 8.0 Hz, 1H) 7.18 (d, *J* = 8.0 Hz, 1H), 6.78 (d, *J* = 8.0 Hz, 1H), 6.70 (s, 1H), 6.65 (d, *J* = 8.0 Hz, 1H), 5.48 (s, 1H), 4.10 (m, 1H), 3.80 (s, 3H), 3.00 (dd, *J* = 4.0, 8.0 Hz, 1H), 2.85 (t, *J* = 8.0 Hz, 2H), 2.60 (t, *J* = 8.0 Hz, 2H), 2.50 (dd, *J* = 2.0, 8.0 Hz, 1H), 1.65 (m, 1H), 1.35 (m, 1H), 1.20 (m, 6H), 0.85 (t, *J* = 8.0 Hz, 3H). ^13^C NMR (CDCl_3_, 100 MHz) *δ* 153.8, 149.2, 146.4, 145.7, 143.9, 132.8, 129.5, 120.8, 118.1, 114.2, 112.1, 110.9, 106.7, 59.1, 55.8, 41.3, 32.5, 32.2, 31.9, 31.5, 24.3, 22.5, 13.9. HRMS-ESI (m/z) [M + H]^+^ calcd for C_23_H_30_N_3_O_4_ 412.2236, found 412.2203.

2-methoxy-4-(2-(1-(4-nitrophenyl)-5-pentyl-4,5-dihydro-1*H*-pyrazol-3-yl)ethyl)phe nol (**5j**). Brown viscous liquid; yield: 78%; R*_f_* = 0.35 (70% dichlorometane: hexane); IR (neat) *υ*_max_ 3433 (OH), 1593 (C=N), 1512 (C=C), 1488 (N-O), 1273 (C-O), 1179 (C-N), 1108 (C-O) cm^−1^. ^1^H NMR (CDCl_3_, 400 MHz) *δ* 8.38 (d, *J* = 8.0 Hz, 2H), 7.18 (d, *J* = 8.0 Hz, 2H), 7.10 (d, *J* = 8.0 Hz, 1H) 7.00 (m, 2H), 5.80 (s, 1H), 4.50 (m, 1H), 4.10 (s, 3H), 3.30 (dd, *J* = 4.0, 8.0 Hz, 1H), 3.18 (t, *J* = 8.0 Hz, 2H), 2.95 (dt, *J* = 2.0, 8.0 Hz, 2H), 2.82 (dd, *J* = 2.0, 8.0 Hz, 1H), 1.95 (m, 1H), 1.65 (m, 1H), 1.50 (m, 6H), 1.15 (t, *J* = 8.0 Hz, 3H). ^13^C NMR (CDCl_3_, 100 MHz) *δ* 156.7, 148.6, 146.4, 144.0, 137.9, 132.5, 126.2, 120.8, 114.3, 110.9, 110.7, 58.4, 55.8, 41.3, 32.4, 32.2, 31.9, 31.4, 24.2, 22.5, 13.9. HRMS-ESI (m/z) [M − H]^+^ calcd for C_23_H_38_N_3_O_4_ 410.2080, found 410.2039.

4-(3-(4-hydroxy-3-methoxyphenethyl)-5-pentyl-4,5-dihydro-1H-pyrazol-1-yl)benzo ic acid (**5k**). Brown viscous liquid; yield: 71%; R*_f_* = 0.40 (2% methanol: dichlorometane); IR (neat) *υ*_max_ 3400 (OH), 1669 (C = O), 1595 (C=N), 1514 (C=C), 1267 (C-O), 1169 (C-N), 1120 (C-O) cm^−1^.^1^H NMR (CDCl_3_, 400 MHz) *δ* 8.20 (d, *J* = 8.0 Hz, 2H), 7.18 (d, *J* = 8.0 Hz, 2H), 7.05 (d, *J* = 8.0 Hz, 1H), 6.95 (m, 2H), 4.40 (m, 1H), 4.05 (s, 3H), 3.20 (dd, *J* = 4.0, 8.0 Hz, 1H), 3.10 (t, *J* = 8.0 Hz, 2H), 2.88 (t, *J* = 8.0 Hz, 2H), 2.72 (dd, *J* = 4.0, 8.0 Hz, 1H), 1.95 (m, 1H), 1.65 (m, 1H), 1.50 (m, 6H), 1.10 (t, *J* = 8.0 Hz, 3H). ^13^C NMR (CDCl_3_, 100 MHz) *δ* 172.3, 154.6, 148.4, 146.4, 143.9, 132.8, 132.0, 120.8, 117.3, 114.3, 111.1, 111.0, 58.4, 55.8, 41.1, 32.5, 32.3, 32.0, 31.5, 24.3, 22.5, 13.9. HRMS-ESI (m/z) [M + H]^+^ calcd for C_24_H_31_N_2_O_4_ 411.2284, found 411.2244.

#### 4.3.3. Synthesis of the Amino Derivatives (**6a**–**6c**, **7**)

To a solution of [6]-shogaol (**4**) (90 mg, 0.32 mmol) in ethanol (5 mL), 2-aminothiophenol (60 mg, 0.48 mmol) was added at room temperature. The mixture was refluxed at 80 °C. After completion based on TLC, the mixture was filtered, and the residue was washed with ethanol (10 mL) and dried with anhydrous sodium sulfate. Evaporation of the combined solvents gave crude product. Purification of the crude product by column chromatography (5% MeOH in CH_2_Cl_2_) gave a yellow viscous liquid of compound **6a** (92 mg, 0.25 mmol, 78%).

10-(4-Hydroxy-3-methoxyphenyl)-6-((2-mercaptophenyl)amino)decan-8-one (**6a**). Yellow viscous liquid; yield: 78%; R*_f_* = 0.55 (100% dichlorometane); IR (neat) *υ*_max_ 3359 (OH), 1710 (C = O), 1512 (C=C), 1267 (C-O) cm^−1^.^1^H NMR (CDCl_3_, 400 MHz) *δ* 7.52 (d, *J* = 8.0 Hz, 1H), 7.36 (t, *J* = 8.0 Hz, 1H), 7.12 (d, *J* = 8.0 Hz, 1H), 7.03 (t, *J* = 8.0 Hz, 1H), 6.85 (d, J = 8.0 Hz, 1H), 6.81 (d, J = 4.0 Hz, 1H), 6.76 (dd, J = 4.0, 8.0 Hz, 1H), 3.84 (s, 3H), 3.71 (m, 1H), 3.04 (m, 2H), 2.87 (m, 2H), 2.39 (dd, *J* = 4.0, 12.0 Hz, 1H), 2.16 (dd, *J* = 4.0, 12.0 Hz, 1H), 1.57 (m, 2H), 1.26 (m, 6H), 0.89 (m, 3H). ^13^C NMR (CDCl_3_, 100 MHz) *δ* 208.3, 146.4, 143.9, 137.5, 135.3, 133.4, 129.4, 124.6, 124.2, 122.7, 120.9, 114.3, 111.2, 57.0, 55.9, 43.0, 39.2, 38.0, 32.1, 29.3, 26.6, 22.5, 14.0. HRMS-ESI (m/z) [M − H_2_O + H]^+^ calcd for C_23_H_30_NO_2_S 384.1997, found 384.1917.

The same procedure was conducted with 2-aminophenol, aniline*,* and *o*-phenylenediamine to convert 6-shogaol (**4**) into **6b**, **6c** and **7**.

10-(4-Hydroxy-3-methoxyphenyl)-6-((2-hydroxyphenyl)amino)decan-8-one (**6b**). Orange viscous liquid; yield: 75%; R*_f_* = 0.40 (100% dichlorometane); IR (neat) *υ*_max_ 3312 (OH), 1709 (C = O), 1508 (C=C), 1269 (C-O) cm^−1^.^1^H NMR (CDCl_3_, 400 MHz) *δ* 7.74 (d, *J* = 8.0 Hz, 1H), 7.39 (m, 3H), 6.78 (d, J = 8.0 Hz, 1H), 6.64 (m, 2H), 6.40 (s, 1H), 6.22 (s, 1H), 5.82 (d, J = 8.0 Hz, 1H), 5.54 (brs, 1H), 3.94 (m, 1H), 3.83 (s, 3H), 2.81 (t, *J* = 8.0 Hz, 2H), 2.71 (m, 3H), 2.59 (dd, *J* = 8.0, 16.0 Hz, 1H), 1.58 (m, 2H), 1.25 (m, 6H), 0.85 (t, *J* = 8.0 Hz, 3H). ^13^C NMR (CDCl_3_, 100 MHz) *δ* 207.7, 146.4, 144.7, 143.9, 142.6, 132.6, 129.0, 128.4, 125.3, 120.8, 116.0, 114.4, 111.0, 55.8, 48.7, 48.9, 45.5, 34.6, 31.5, 29.3, 25.8, 22.5, 14.0. HRMS-ESI (m/z) [M + 2H]^+^ calcd for C_23_H_33_NO_4_ 387.2410, found 387.2668.

10-(4-Hydroxy-3-methoxyphenyl)-6-(phenylamino)decan-8-one (**6c**). Yellow viscous liquid; yield: 74%; R*_f_* = 0.65 (100% dichloro-metane); IR (neat) *υ*_max_ 3390 (OH), 1705 (C = O), 1512 (C=C), 1266 (C-O) cm^−1^.^1^H NMR (CDCl_3_, 400 MHz) *δ* 7.17 (d, *J* = 8.0 Hz, 1H), 7.16 (d, *J* = 8.0 Hz, 1H), 6.81 (d, J = 8.0 Hz, 1H), 6.69 (m, 1H), 6.64 (s, 1H), 6.63 (m, 1H),6.58 (m, 1H), 3.85 (s, 3H), 3.84 (m, 1H), 2.79 (t, *J* = 8.0 Hz, 2H), 2.68 (t, *J* = 8.0 Hz, 2H), 2.63 (m, 1H), 2.55 (dd, *J* = 8.0, 16.0 Hz, 1H), 1.51 (m, 2H), 1.26 (m, 6H), 0.88 (m, 3H). ^13^C NMR (CDCl_3_, 100 MHz) *δ* 209.6, 147.2, 146.4, 143.9, 132.9, 129.4, 120.7, 117.4, 114.3, 113.4, 111.0, 55.8, 49.9, 47.5, 45.6, 35.3, 31.9, 29.3, 26.3, 22.7, 14.1. HRMS-ESI (m/z) [M + H]^+^ calcd for C_23_H_32_NO_3_ 370.2382, found 370.2385.

2-Methoxy-4-(2-(2-pentyl-2,3-dihydro-1*H*-benzo[*b*][1,4]diazepin-4-yl)ethyl)phenol (**7**). Yellow viscous liquid; yield: 70%; R*_f_* = 0.45 (100% dichlorometane); IR (neat) *υ*_max_ 3727 (NH), 3292 (OH), 1704 (C = O), 1512 (C=C), 1230 (C-O) cm^−1^.^1^H NMR (CDCl_3_, 400 MHz) *δ* 7.17 (dd, *J* = 4.0, 8.0 Hz, 1H), 6.97 (m, 2H), 6.84 (d, *J* = 8.0 Hz, 1H), 6.80 (d, *J* = 4.0 Hz, 1H), 6.73 (m, 2H), 3.85 (s, 3H), 3.84 (m, 1H), 2.98 (t, *J* = 8.0 Hz, 2H), 2.82 (t, *J* = 8.0 Hz, 2H), 2.41 (dd, *J* = 4.0, 12.0 Hz, 1H), 2.22 (dd, *J* = 8.0, 16.0 Hz, 1H), 1.52 (m, 2H), 1.31 (m, 6H), 0.89 (t, *J* = 8.0 Hz, 3H). ^13^C NMR (CDCl_3_, 100 MHz) *δ* 174.1, 146.4, 143.9, 139.3, 138.1, 133.5, 127.6, 125.7, 121.3, 121.0, 114.3, 114.3, 111.2, 65.7, 55.9, 44.0, 38.3, 37.7, 32.2, 31.7, 25.6, 22.6, 14.0. HRMS-ESI (m/z) [M + H]^+^ calcd for C_23_H_31_N_2_O_2_ 367.2386, found 367.2385.

### 4.4. HDAC Activity Assay

The semi-synthetic derivatives were evaluated for their ability to inhibit HDAC enzymes. Inhibition of HDAC activity in vitro was assessed using the Fluor-de-Lys HDAC activity assay kit (Biomol, Enzo Life Sciences International, Inc., Farmingdale, NY, USA). The HeLa nuclear extract provided with the kit was used as a source of HDAC enzymes for the in vitro study. TSA was used as the positive control. The HeLa nuclear extract, substrate, buffer and inhibitors were incubated. Deacetylation of the substrate was performed next by adding a developer to generate a fluorophore. The spectra Max Gemini XPS microplate spectrofluorometer (Molecular Devices, San Jose, CA, USA) was used to measure fluorescence signal with excitation at 360 nm and emission at 460 nm. A decrease in fluorescence signal indicated an inhibition of HDAC activity. Trichostatin A (TSA) was used as a positive control. All experiments were carried out in triplicate.

### 4.5. MTT Assay

The MTT reduction assay was performed with non-cancer (Vero), human cervical cancer (HeLa), human colon cancer (HCT116), human breast adenocarcinoma cancer (MCF-7), and cholangiocarcinoma (H-69 (Vero), KKU-100 and KKU-M213B) cell lines according to the previously described method [34,35]. Briefly, cells were seeded in 96-well plates. The next day, cells were exposed to the selected compounds at various concentrations and incubated for 24, 48 and 72 h. After incubation, the culture medium was exchanged with 110 µL of MTT (0.5 mg/mL in PBS medium) and further incubated for 2 h. The amount of MTT formazan product was determined after dissolving in DMSO by measuring its absorbance with a microplate reader (Bio-Rad Laboratories, Hercules, CA, USA) at a test wavelength of 550 nm and a reference wavelength of 655 nm. The cell viability was expressed as a percentage of the viable cells of control culture conditions, and the IC_50_ values of each group were calculated.

### 4.6. Molecular Docking Studies

The crystal structures of HDAC1, HDAC2, HDAC3 and HDAC8 (PDB entry code: 4BKX, 3MAX, 4A69 and 1T64, respectively) were obtained from the Protein Data Bank (http://www.rcsb.org/pdb (accessed on 1 March 2022). All water and non-interacting ions as well as ligands were removed. Then, all missing hydrogen and side-chain atoms were added using the ADT program. Gasteiger charges were calculated for the system. For ligand setup, the molecular modeling program Gaussview was used to build the ligands. These ligands were optimized with the AM1 level by using Gaussian03W. Molecular docking studies were performed for 50 runs using AutoDockTools 1.5.4 (ADT) (La Jolla, CA, USA) and AutoDock 4.2 programs and Lamarckian genetic algorithm search. A grid box size of 60 × 60 × 60 points with a spacing of 0.375 Å between the grid points was implemented and covered almost the entire HDAC protein surface. For TSA and other inhibitors, the single bonds were treated as active torsional bonds.

## Data Availability

The data presented in this study are available on request from the corresponding author.

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
