# Peer review of "Histone Deacetylase Inhibitory Activity and Antiproliferative Potential of New [6]-Shogaol Derivatives"

_molecules, 2022, doi:10.3390/molecules27103332_

Round 1

Reviewer 1 Report

    In the paper entitled “Histone Deacetylase Inhibitory Activity and Antiproliferative Potential of New [6]-Shogaol Derivatives”, 20 novel [6]-shogaol derivatives were synthesized and structurally characterized by FT-IR, 1H-NMR, 13C-NMR, and HR-ESI-MS. Besides, the inhibitory activities of the title compounds against histone deacetylases (HDACs, including HDAC1, HDAC2, HDAC3 and HDAC8) were evaluated, and it was found that all of them showed moderate to good histone deacetylase inhibition at 100 μM. The interactions of the most potent derivatives 4c, 4d, 5j and 5k with HDACs were further investigated by molecular docking, and their antiproliferative activities against non-cancer (Vero), human cervical cancer (HeLa), human colon cancer (HCT-116), human breast adenocarcinoma cancer (MCF-7), and cholangiocarcinoma (KKU100 and KKU-M213B) cells were tested by MTT assay. This work could provide a theoretical basis for the discovery and development of novel HDACs inhibitors. I recommend this manuscript be accepted after major revision.

1. The introduction section seems to be too simple, and thus should be re-written. A lot of key informations should be indicated in this section. For example, what are the sources and research advances of the natural compound [6]-shogaol? Why the authors chose it to be the leading compound for further chemical modification with the aim of developing HDACs inhibitors? Similarly, what were the reasons for introducing hydrazone, pyrazole, amino and imine moiety into the molecular skeleton of [6]-shogaol? And, what was the design strategy of the title compounds?

2. Please provide the synthetic routes of the title compounds and related discussion about chemistry in this manuscript.

3. The inhibitory activities of all the title compounds against HDACs were evaluated, for preliminary screening of potent compounds. Based on the results, four representative compounds 4c, 4d, 5j and 5k were selected for further antiproliferative activity evaluation and molecular docking. So, what was the relationship of their HDACs inhibition activity and the antiproliferative activity in this work? And, what was the definition of the ΔG and Ki in Table 2? What was the correlation between the in silico binding affinities of the selected compounds and the experimental ones?

4. How the authors could assign every C atom of the title compounds for the characteristic signals in the 13C-NMR of the corresponding compound?

5. There are still a few errors. Please check carefully for the whole text.

Line 19 in page 1: “100 M” should be “100 μM”;

Lines 111 and 115 in page 3: “¶-¶” should be “π-π”.

Author Response

Dear  Reviewer,

            Thank you very much for reviewing our manuscript. Here are our revision and response.

Major Revisions:

Response to reviewer #1.

  1. The introduction section seems to be too simple, and thus should be re-written. A lot of key informations should be indicated in this section. For example, what are the sources and research advances of the natural compound [6]-shogaol? Why the authors chose it to be the leading compound for further chemical modification with the aim of developing HDACs inhibitors? Similarly, what were the reasons for introducing hydrazone, pyrazole, amino and imine moiety into the molecular skeleton of [6]-shogaol? And, what was the design strategy of the title compounds?

Authors reply: All changes of the introduction were highlighted in red (page 2, lines 62-71).

  1. Please provide the synthetic routes of the title compounds and related discussion about chemistry in this manuscript.

Authors reply: The synthetic routes of the title compounds were provided in Scheme 1. The related discussion about chemistry was added (page 2-3, lines 95-103).

  1. The inhibitory activities of all the title compounds against HDACs were evaluated, for preliminary screening of potent compounds. Based on the results, four representative compounds 4c, 4d, 5j and 5k were selected for further antiproliferative activity evaluation and molecular docking. So, what was the relationship of their HDACs inhibition activity and the antiproliferative activity in this work?

Authors reply: Anticancer activity of the best-four HDAC inhibitors were investigated on proliferation of five human cancer cell lines in a time-dependent manner. All selected compounds showed antiproliferative activity on growth inhibition of cancer cell lines. The results support potential anticancer candidates of these HDAC inhibitors (page 5, lines 170-173).

  1. What was the definition of the ΔG and Ki in Table 2?

 Authors reply: Free energy of binding (ΔG, kcal/mol) and inhibition constant (Ki, µM) were showed in the bottom of Table 2.

  1. What was the correlation between the in silico binding affinities of the selected compounds and the experimental ones?

Authors reply: The in vitro HDAC inhibitions of the obtained compounds were carried out with the assay kit containing mixed-HDAC isoforms. In order to further investigate isoform selectivity of top-four potent HDAC inhibitors, the in silico experiment was performed with each HDAC isoform. Therefore, both results were not fully related. However, compounds with good binding affinities should have a potential to be specific HDAC inhibitor for each isoform (page 5, lines 150-155).

  1. How the authors could assign every C atom of the title compounds for the characteristic signals in the 13C-NMR of the corresponding compound?

Authors reply: We originally assigned the C atom by comparing with the previously published data of [6]-shogaol (ref: 27). However, the complete 2D-NMR data were not obtained. Thus, the numbering of C atom was removed in the revised manuscript. 

  1. Line 19 in page 1: “100 M” should be “100 μM”; Lines 111 and 115 in page 3: “¶-¶” should be “π-π”.

Authors reply: All changes were highlighted in red.

.  

Reviewer 2 Report

It is a work that from the chemical point of view does not have significant contributions, however, it is interesting from the biological point of view, which makes it appropriate to be published in Molecules. The way the values for the C13 are reported does not follow the guidelines, it should be reported with a decimal, the reference 4 and some commas should be reviewed

Author Response

Dear Reviewer,

            Thank you very much for reviewing our manuscript. Here are our revision and response.

Response to reviewer #2.

  1. The way the values for the C13 are reported does not follow the guidelines, it should be reported with a decimal, the reference 4 and some commas should be reviewed.

Authors reply: All C13 values were changed as a decimal. All references were checked.  

Round 2

Reviewer 1 Report

Th revised manuscript is OK.